# Investigating the presence of adsorbed species on Pt steps at low potentials

Rubén Rizo[1✉], Julia Fernández-Vidal[2], Laurence J. Hardwick[2], Gary A. Attard[3], Francisco J. Vidal-Iglesias [1], Victor Climent [1], Enrique Herrero [1✉] & Juan M. Feliu [1✉]

The study of the OH adsorption process on Pt single crystals is of paramount importance since this adsorbed species is considered the main intermediate in many electrochemical reactions of interest, in particular, those oxidation reactions that require a source of oxygen. So far, it is frequently assumed that the OH adsorption on Pt only takes place at potentials higher than 0.55 V (versus the reversible hydrogen electrode), regardless of the Pt surface structure. However, by CO displacement experiments, alternating current voltammetry, and Raman spectroscopy, we demonstrate here that OH is adsorbed at more negative potentials on the low coordinated Pt atoms, the Pt steps. This finding opens a new door in the mechanistic study of many relevant electrochemical reactions, leading to a better understanding that, ultimately, can be essential to reach the final goal of obtaining improved catalysts for electrochemical applications of technological interest.

[1] Instituto de Electroquímica, Universidad de Alicante, Apdo. 99, E-03080 Alicante, Spain. [2] Stephenson Institute for Renewable Energy, University of Liverpool, Peach Street, Liverpool L69 7ZF, UK. [3] Department of Physics, University of Liverpool, Crown Street, Liverpool L69 7ZD, UK. ✉email: ruben.rizo@ua.es; herrero@ua.es; juan.feliu@ua.es

Platinum is, by far, the most frequently employed metal as electrode material in electrocatalysis due to its excellent electrocatalytic properties. Furthermore, most of the electrocatalytic processes are surface-sensitive reactions. The use of well-defined surfaces, namely, single crystal electrodes, has allowed studying in detail such dependencies between reactivity and surface structure[1,2]. Moreover, the nature of the species adsorbed on the electrode surface has been revealed to exert a significant impact on the electrocatalytic response[3–6]. For the Pt(111) surface, the voltammogram recorded in the absence of specifically adsorbed anions shows two well-defined regions separated by the double layer area, which were both originally attributed to hydrogen adsorption[7,8]. However, in the immediate years that followed the initial report of the characteristic voltammogram of this surface, controversy arose about the nature of the voltammetric currents above the double layer region[9–11]. In the following years, CO displacement experiments demonstrated that the adsorption state at potentials higher than 0.5 V[12,13] is dominated by anion adsorption. For these experiments, CO is dosed inside the electrochemical cell at a given constant potential, at which CO is readily adsorbed on the surface displacing the species initially present on the Pt surface. Simultaneously, the transient current required during the forced desorption process to maintain the imposed potential value is recorded[14]. During this process, since there is no faradic flow of charged species through the interphase other than the displaced charge, the charge recorded under the transient current curve ($q_{dis}$) corresponds to the difference between the total electrode charge before ($q_i$) and after ($q_f$) CO adsorption:

$$q_{dis} = q_f - q_i \qquad (1)$$

The initial charge ($q_i$) is the charge of interest since it corresponds to the charge on the Pt surface in absence of CO at the potential of the experiment. To calculate $q_i$, the final charge ($q_f$) after CO adsorption is necessary. $q_f$ can be calculated from the integration of the differential capacity of the Pt surface covered with CO between the potential of the measurement and the estimated potential of zero charge (pzc) of the CO-saturated surface. Determination of the latter is not a straightforward process[15,16]. However, due to the small differential capacity of the CO-covered Pt surfaces in comparison with that recorded in absence of CO, $q_f$ can be considered negligible as a first

approximation. For that reason, the $q_{dis}$ can be taken as follows:

$$q_{dis} \approx -q_i \qquad (2)$$

Thus, a measurement of the total charge on the electrode at a certain potential can be directly estimated from the value of the charge displaced by CO. In addition, by carrying out the CO displacement experiments at different potentials in an interval where $CO_{ads}$ is not oxidized, the curve $q$ versus $E$ can be built. This curve must be the same as the curve obtained by integrating the voltammetric current when the processes occurring are reversible enough to be considered at equilibrium at the recording scan rate:

$$q(E) = \int_{E^*}^{E} \frac{j}{v} dE + q(E^*) \qquad (3)$$

where $j$ is the voltammetric current density and $v$ is the sweep rate. The integration constant, $q(E^*)$ can be determined from the charge displaced at the origin of the integration curve, $-q_{dis}(E^*)$. Once the curve $q(E)$ has been obtained, the potential of zero total charge (pztc) can be directly measured from the intersection of this curve with the axis of abscissas. Additional explanations about the calculations can be found in the supporting information. The pzc is a key parameter in electrochemistry, needed to understand the structure of the interphase, including the orientation of solvent molecules and the adsorption of ionic species. Thus, when positive current transients are recorded during CO displacement (which imply that the surface has a negative total charge before the introduction of CO), the main process involved is, clearly, the oxidative displacement of adsorbed hydrogen, according to:

$$Pt - H + CO \rightarrow Pt - CO + H^+ + e^- \qquad (4)$$

In absence of specific adsorption of anions, adsorbed oxygenated species, namely OH, coming from water adsorption must be considered the main reason for obtaining negative current during the displacement reaction[13]:

$$Pt - OH + CO + e^- \rightarrow Pt - CO + OH^- \qquad (5)$$

The total charge curve resulting from the combination of the CO displacement experiments carried out on the Pt(111) electrode, and the integration of the voltammetric response shows that the total charge at $E > 0.55$ V is positive (Fig. 1, red curve). In fact, when iodine was used as displacement agent, a negative displaced charge was obtained[17], as expected and thereby demonstrating the presence of adsorbed OH at this potential region[12]. Figure 1 shows the actual state of the art of the different adsorption states of a well-ordered Pt(111) surface recorded in 0.1 M HClO₄ solution at a scan rate of 50 mV s⁻¹, together with the total charge curve obtained using Eq. (3) As mentioned above, two different regions can be clearly distinguished: hydrogen adsorption/desorption at potentials lower than 0.40 V and the OH adsorption region, which occurs at potentials higher than 0.55 V[12].

Conversely, the controlled introduction of monoatomic steps on the well-ordered Pt(111) surface, by cutting the crystal in a tilted orientation, leads to the appearance of peaks in the hydrogen adsorption/desorption potential region of the cyclic voltammogram. The position of these peaks depends on the geometry of the step, pH, and nature of the cation in solution and its charge increases with the step density[18–21]. It has been generally assumed that the hydrogen adsorption/desorption process is responsible for this peak, although some results suggest that OH adsorption can also be involved in these processes[22–24]. Indeed, DFT results suggest that cation coadsorption with OH is responsible for the observed voltammetric behaviour[25,26]. This fact has not yet been demonstrated spectroscopically and the

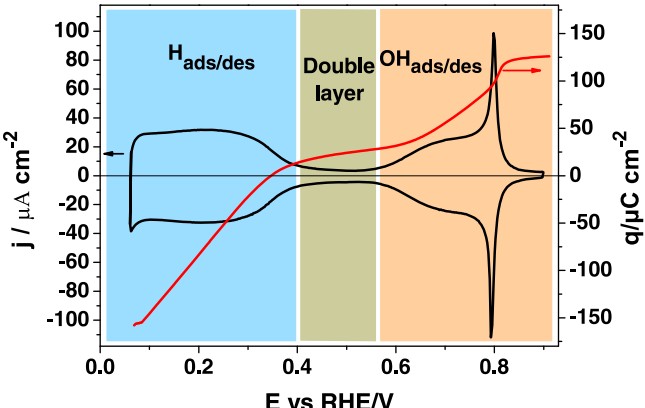

**Fig. 1 State of the art for Pt(111).** Cyclic voltammogram (black line, left hand axis) and charge density curve (red line, right hand axis) for Pt (111) recorded in 0.1 M HClO₄ solution at a scan rate of 50 mV s⁻¹. The region in blue corresponds to the hydrogen adsorption/desorption region, the region in green to the double layer, and the region in orange to the hydroxyl adsorption/desorption process.

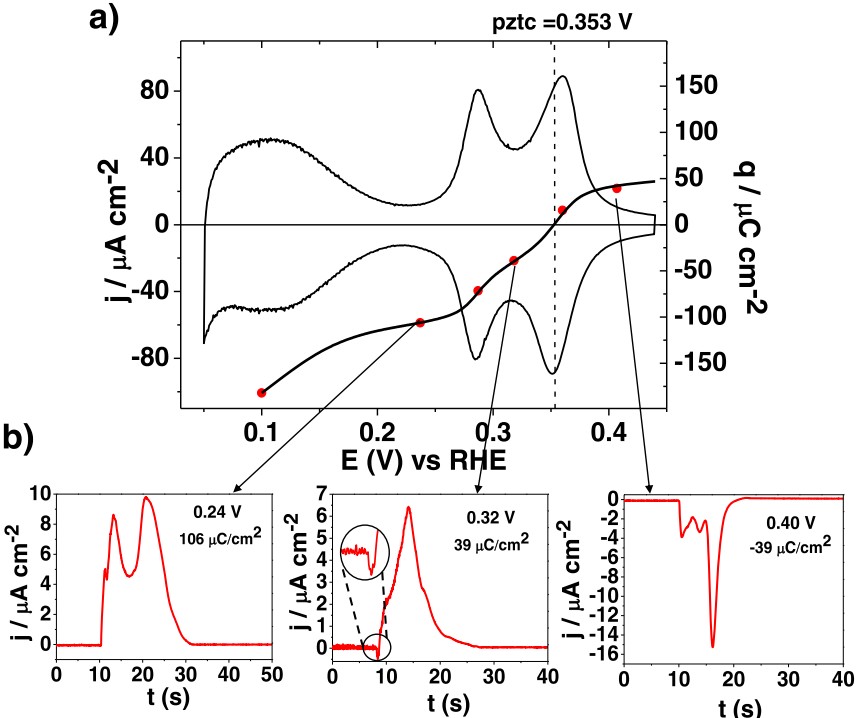

**Fig. 2 CO displacement experiments on Pt(311). a** Cyclic voltammogram and total charge curve for Pt(311) surface as a function of the potential recorded at a scan rate of 50 mV s⁻¹ and **b** CO displacement experiments at different potentials in 0.1 M HClO₄ solution.

presence of OH adsorbed on the steps in the hydrogen adsorption/desorption region is a question that needs to be clarified. Most of the oxidation reactions of organic molecules and even oxygen reduction reaction, which are of paramount importance for fuel cells and batteries, require the presence of adsorbed $OH_{ads}$[1,6,27]. Moreover, these peaks observed in the voltammogram of stepped surfaces have a clear counterpart in similar peaks observed for polycrystalline or nanoparticle samples, which are of practical interest, stressing the importance of understanding this issue in the behavior of platinum as an electrocatalyst.

In this work, we carry out CO displacement experiments on stepped surfaces. The analysis of the data, together with the observed behavior of the alternating potential voltammetry (AC voltammetry) and the bands identified by using Raman spectroscopy report on the presence of adsorbed OH on the step sites at low potentials.

## Results and discussion
The $H_{ads}$ and $OH_{ads}$ electrosorption processes on platinum are sensitive to the crystallographic structure of the electrode since the adsorption energies for both species depend on the geometry of the particular adsorption site[22]. In this sense, the voltammetric contributions of the (110) steps on the (111) terraces appear at lower potential values than those corresponding to the (100) steps. Here, since the main purpose of this work is to investigate the adsorption of species on the step separated from the contribution of the terrace, the Pt(311) surface was selected: on this surface, which is composed of two-atoms wide (111) terraces separated by monoatomic (100) steps, the voltammetric contributions of the (100) steps are almost detached from those arising from the terraces in 0.1 M HClO₄ (Fig. 2a). It is worth pointing out that this surface can equally be considered as composed of two rows wide (100) terraces separated by monoatomic (111) steps. The voltammetric fingerprint of this surface is characterized by three different signals between 0.06 and 0.4

V[28–30]. The broad signal between 0.06 and 0.2 V corresponds to the adsorption of hydrogen on the (111) terrace, whereas the two peaks at 0.28 and 0.36 V are related to adsorption processes on the (100) steps. It should be noted that, for the voltammograms of surfaces in the n(111) × (100) zone with short terraces, the step contributions split into two separate peaks, whereas for long terraces only one peak at ca. 0.32 V is obtained.

On this electrode, CO displacement experiments have been performed at different potentials, and some of the current transients are shown in Fig. 2b. Voltammograms recorded before and after the CO displacement experiment were identical (Fig. S1) which assures that the surface structure of the electrode is maintained during the experiment. The total charge curve as a function of the potential has been calculated by integrating the voltammetric current, using the value of the charge displaced at 0.10 V as the integration constant ($q(E^*)$ in Eq. (3)) (Fig. 2a, black line)[14]. The potential of zero total charge (pztc) can be directly measured from the intersection of this curve with the axis of abscissas and, for this electrode, the measured pztc is at 0.353 ± 0.005 V. This value is very close to the peak potential of the second signal related to the step. Additionally, the measured charge during the CO displacement current transients (after changing its sign) at different potentials was plotted together with the curve obtained from the integration of the voltammogram (red dots, Fig. 2a). As expected, the values of the charge displaced at the different potentials overlap with the integrated curve, which validates the use of the CO displacement charge to calculate the total charge on the electrode. These measurements also prove that, in the potential range of study, the involved processes ($H_{ads}$ and $OH_{ads}$) are reversible enough to be considered at equilibrium.

The analysis of the transients can also shed light on the nature of the species adsorbed. When the CO displacement is carried out at potentials more negative than the onset for the first peak related to the step (below 0.24 V), the currents recorded are positive due to the displacement of hydrogen, following reaction

1. The current drops to zero when the surface is totally covered by CO. However, at 0.32 V, a potential which is located between the two peaks, the curve for the transient current shows a bipolar shape, with negative values during the initial time ($t \leq 1s$), and after that, the current becomes positive. This bipolar, non-monotonic nature suggests that two different species are adsorbed on the surface, namely $H_{ads}$ and $OH_{ads}$. This bipolar shape was consistently observed in all the displacement experiments carried out at this potential, while it was consistently absent at the other potentials studied. The simplest explanation for this result is that, during the initial stages of the CO adsorption experiment, mainly $OH_{ads}$ is displaced from the surface, giving rise to the negative current according to Eq. (2). At longer times, the displacement of $H_{ads}$ yields a positive net current and because the net integrated charge is positive, the hydrogen coverage must be larger than the OH coverage. The presence of $OH_{ads}$ on the steps is confirmed by the negative current recorded in the transient curve at 0.40 V. It should be stressed that, when the same experiment is carried out on a flat Pt(111) surface, nearly zero charge is displaced at this potential.

In order to estimate the $OH_{ads}$ and $H_{ads}$ on the step, a charge analysis has to be carried out (Note 1, Fig. S2). In this analysis, it will be assumed that at 0.24 V, just before the onset of the peak with the step contributions, only $H_{ads}$ is present on the surface. Moreover, at 0.24 V, $H_{ads}$ from the terrace has been almost completely desorbed and thus, $q_i$ at this potential (-106 μC cm$^{-2}$) should correspond to the charge related to $H_{ads}$ on the step. The theoretical maximum charge for $H_{ads}$ on the step assuming that one hydrogen atom is absorbed per Pt step atom can be calculated according to the hard-sphere model[31]. For the surfaces having $n$-atom wide (111) terraces and monoatomic (100) steps (whose Miller index are Pt($n+1$,$n$-1,$n$-1)) the charge on the steps is given by the expression:

$$q^{step}_{(n+1, n-1, n-1)} = \frac{q_{Pt(111)}}{n - 1/3} \cos(\alpha) \qquad (6)$$

where $q_{Pt(111)}$ is the charge density measured for a process taking place on a Pt(111) electrode transferring 1e$^-$ per platinum site (which stands for 241 μC cm$^{-2}$) and $\alpha$ is the angle between the surface and the (111) plane. For the Pt(311) surface, $n$ is equal to 2, $\cos(\alpha)$ is 0.870, and the calculated charge for the step with a $H_{ads}$ coverage equal to 1 is 125.9 μC cm$^{-2}$. This value implies that the calculated $H_{ads}$ coverage on the step is ca. 0.84. This calculation has considered that $q_f$, the charge after CO is displaced, is negligible. To estimate this charge, it would be necessary to know the potential of zero charge of the electrode covered by CO, or at least, the work function of this surface covered by CO, but this data is not available for this surface. For the CO covered Pt(111), the estimation of the potential of zero charge gives a value of ca. 1.1 V and a value of $q_f$ of ca. −13 μC cm$^{-2}$ [15,32]. For the Pt(311), the expected diminution of the work function with respect to the Pt(111) surface due to the presence of steps[33] should give a lower value of $q_f$. Taking −10 μC cm$^{-2}$ as a reasonable value for $q_f$, $q_i$ would be then around −116 μC cm$^{-2}$, which gives a hydrogen coverage close to 1 (ca. 0.92).

Alternatively, the value of $q_i$ at 0.40 V, just after the second peak, should correspond to the charge related to the displacement of $OH_{ads}$ from the step (39 μC cm$^{-2}$), and thus the OH coverage would stand for 0.31. If the value of $q_{dis}$ is corrected with that of $q_f$ at this potential (ca. -9 μC cm$^{-2}$), the OH coverage is 0.24. These values imply that the $OH_{ads}$ coverage on the step is close to ¼ whereas that of $H_{ads}$ is close to 1.

The charge analysis shows that $H_{ads}$ and $OH_{ads}$ are involved in the voltammetric peaks between 0.25 and 0.4 V. It would be tempting to assign the first peak to $H_{ads}$ and the second peak to $OH_{ads}$, however, the previous analysis shows that this cannot be the case since the charges under the peaks (67 μC cm$^{-2}$ for the peak at 0.28 V and 78 μC cm$^{-2}$ for that at 0.36 V) do not match those

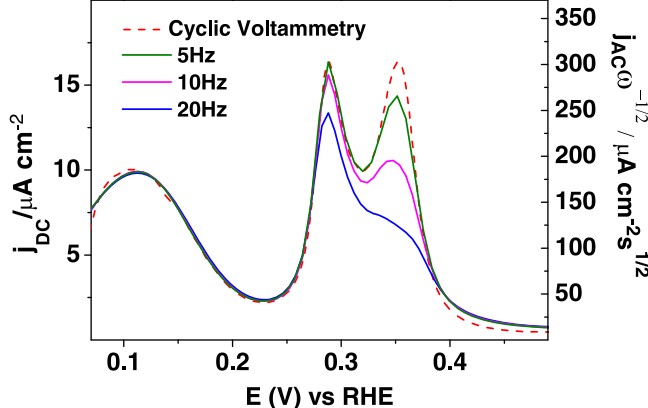

**Fig. 3 AC voltammetry on Pt(311).** AC voltammetry and cyclic voltammetry for different frequencies on a Pt(311) at 10 mV/s.

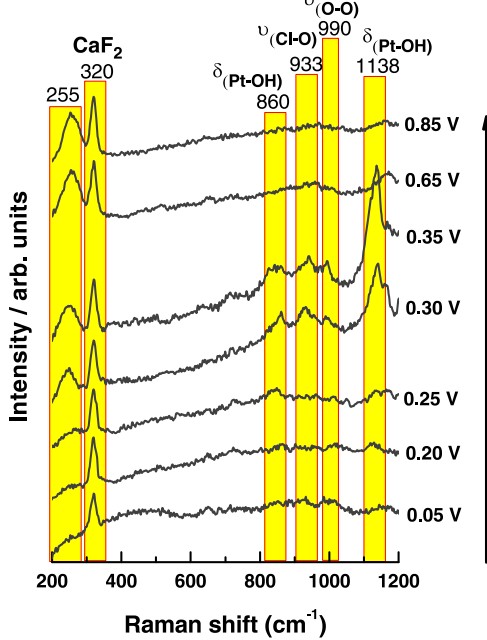

**Fig. 4 Raman spectra of the hydroxyl phase at low potentials on Pt(311).** Potential dependent in situ SHINERS spectra of Pt(311) electrode in 0.1 M HClO₄ electrolyte purged with Ar. The different Raman bands are highlighted by the yellow squares.

corresponding to the individual processes. Thus, they contain mixed contributions from both and cannot be distinguished under equilibrium conditions. The only chance to separate both processes is if they had different time constant and responded differently to a fast perturbation of the potential. Alternating current (AC) voltammetry can be used for this purpose, and in fact, the same approach was used in the past to discriminate between different adsorption processes during the oxidation of Pt(111)[34]. Since the AC current measured in the AC voltammogram depends on the reaction rate, it is expected that two different adsorption processes can be distinguished by changing the frequency ($\omega$) of the sinusoidal wave superimposed to the linear sweep. Figure 3 shows the AC voltammograms obtained for different frequencies and their comparison with the linear voltammetry.

Since the AC current for fast processes scales up with the square root of the frequency, AC currents have been divided by $\sqrt{\omega}$ for comparison. As can be seen, normalized AC currents for $H_{ads}$ on the terrace sites are independent of $\omega$ and the shape is exactly the same as

that obtained in the cyclic voltammetric experiment. This indicates that this process is sufficiently fast to respond to the periodic perturbation. In fact, in acidic solutions, impedance measurements on Pt(111) electrodes cannot discriminate this process from that of the double layer charging, which implies its rate cannot be measured using standard procedures[35,36]. However, for the peaks related to the adsorption processes on the steps, changes are observed when $\omega$ increases. As shown in Fig. 3, for $\omega = 20$ Hz, the currents for the second peak significantly diminish, indicating that this peak contains contributions that have a lower adsorption rate, namely $OH_{ads}$. However, the charge under this peak (78 $\mu$C cm$^{-2}$) is larger than the charge for the OH adsorption process (39 $\mu$C cm$^{-2}$) which implies that this peak has contributions from both $H_{ads}$ and $OH_{ads}$. On the other hand, the peak at 0.28 V contains mainly contributions from $H_{ads}$. This result agrees with the observed bipolar transient for the CO displacement in the region between the two peaks.

To further confirm the presence of OH adsorbed on the steps, electrochemical shell-isolated nanoparticle enhanced Raman spectroscopy (EC-SHINERS) was performed. Figure 4 shows representative surface-enhanced Raman spectra at various recorded potentials. As expected from the CO displacement experiments displayed in Fig. 2, electrochemical SHINERS results (Fig. 4) show no significant Raman peaks at potentials below 0.25 V other than the CaF$_2$ band from the Raman window, suggesting that only $H_{ads}$ is present on the surface at these potentials. Above 0.25 V, new Raman bands arise at 255, 860, 933, 990, and 1138 cm$^{-1}$. Despite purging the measurement cell and electrolyte with argon, trace O$_2$ cannot be fully discounted. In this regard, based on previous DFT studies on Pt(111) surfaces, the band at 990 cm$^{-1}$ can be correlated to O$_2$ species bonded to the Pt(311) surface in the presence of OH, corresponding to $\upsilon_{Pt-O-O}$ for the $[Pt(311)OH_2-O_2]^+$ system[37]. Moreover, DFT calculations showed that, while the Pt-OH bending mode $\delta_{PtOH}$ on Pt(100) sites is around 881 cm$^{-1}$, the presence of an atomic oxygen nearby causes a shift towards higher wavenumbers in the Pt-OH vibration due to the constructive role that the oxygen atom plays in bending the H atom[37]. Thus, the signal at 860 cm$^{-1}$ is assigned to Pt-OH bending vibrations from OH adsorbed onto (100) steps near Pt atoms not coordinated to atomic oxygen while the signal at 1138 cm$^{-1}$ can be ascribed to OH adsorbed on Pt atoms surrounded by Pt sites bonded to oxygen. The small deviations in wavenumber position concerning the theoretical values can be attributed to the effect of energy perturbation due to the presence of the step on the OH bending mode.

The intensity of the bands at 860, 933, and 1138 cm$^{-1}$ increase with the applied potential and then disappear above 0.65 V. The potential dependence of the band intensity and the correlation of the onset potential with the voltammetric signal for the steps (0.25 V) reveal that these contributions must be attributed to adsorbed OH, generated by water dissociation on the steps and not to oxygenated products formed from the reduction of the traces of oxygen in the electrolyte[6,28]. If that were the case, and considering that the onset potential for oxygen reduction reaction (ORR) is around 0.9 V for Pt(311) surface, these bands should increase by decreasing the applied potential with no dependence on the peak for the step. Further confirmation of the presence of OH adsorbed at the step is the band at 933 cm$^{-1}$, assigned to the symmetric stretching mode of the perchlorate ion ($\nu_{s(ClO4-)}$). Koper et al.[37] proposed that the formation of OH on Pt (111) is associated with a specific interaction of $ClO_4^-$ with the $OH_{ads}$ layer, which gives rise to the band in the spectra:

$$Pt(111) + H_2O + ClO_4^- \rightleftarrows Pt(111) - OH \cdots ClO_4^- + H^+ + e^- \tag{7}$$

The onset for the band at 933 cm$^{-1}$ is 0.30 V (Fig. 4) and becomes more intense at 0.35 V, coinciding with the voltammetric

peak for the steps, which suggest that, in analogy with the findings of Koper et al., the interaction of $ClO_4^-$ with the $OH_{ads}$ on the steps is responsible for this emerging band, which disappears by increasing the potential above 0.65 V (Fig. S3). This type of specific interaction between the Pt-OH surface and perchlorate anion is found to inhibit the ORR suggesting a direct relationship between surface and reactivity[38]. Finally, the band at 255 cm$^{-1}$ shows an increasing tendency with the applied potential within the whole potential range. This feature is not easily explained by either of the oxygenated adsorbates described below. Koper et al, assigned bands at similar energies to a superoxide species formed at potentials above 1 V for Pt(111) surface. However, the oxidation of the surface in our working potential range is discarded. Another possibility involves the formation of superoxide species from the reduction of possible traces of oxygen in the solution, however, an increase in this signal with the potential in the opposite direction would be expected. Therefore, the signal at 255 cm$^{-1}$ may be tentatively attributed to a vibration of the Si-O bond[39] from the SiO$_2$-coated SHINERS employed for the in situ Raman experiments, which may experience structural changes induced by the effect of the electric field caused by applying positive potentials. Proposed band assignment and adsorption mechanisms discussed above for potentials above 0.25 V for Pt (311) are shown in Fig. S4.

In summary, the existence of $OH_{ads}$ on (100) steps at low potentials and its contribution to the voltammetric peaks that appear at ca. 0.25–0.45 V has been experimentally confirmed, thereby resolving a long-speculated question about the electrochemical reactivity of Pt surfaces and the chemical identity of the present adsorbed species. The CO displacement experiments show negative currents at potentials corresponding to the steps from which an $OH_{ads}$ coverage close to 0.24 was calculated. Further confirmation of the presence of $OH_{ads}$ was obtained by AC voltammetry, showing a perturbation on the shape of the profile by increasing the frequency and confirming the presence of at least two processes with different response times. Furthermore, the presence of $OH_{ads}$ was directly identified by using SHINERS. The presence of $OH_{ads}$ species on Pt surfaces crucially sheds light on a variety of reaction mechanisms whereby $OH_{ads}$ is considered the main intermediate. Further studies, using different Pt surface orientations, are in progress to further study the effect of surface geometry on the adsorption of OH.

## Methods

**Chemicals**. Chloroauric acid (99.9%), (3-aminopropyl)trimethoxysilane (APTMS) (97%), sodium citrate (99.0%), and sodium silicate solution (27% SiO$_2$), employed for the preparation of the Au nanoparticle coated by a thin SiO$_2$ layer (Au@SiO$_2$ NPs), were purchased from Sigma Aldrich. For the electrochemical experiments, solutions were prepared using HClO$_4$ (Merck, Suprapur, 70%) dissolved in ultra-pure water (Elga PureLab Ultra, 18.2 M$\Omega$ cm) and deoxygenated using Ar (N50, Air Liquide). H$_2$ (N50, Air Liquide) for the reference electrode and CO (N50, Air Liquide) for the CO displacement experiments were also employed.

**Electrochemistry**. The electrochemical experiments were carried out in a two-compartment electrochemical glass cell for the CO displacement and AC Voltammetry and a Teflon cell for the Raman experiments. A reversible hydrogen electrode (RHE) was employed as reference electrode and a Pt wire as counter electrode. The Pt(311) working electrode was prepared by following the Clavilier method[2]. The electrolyte (0.1 M HClO$_4$) was always deaerated with Ar before experiments. CO is introduced in the deaerated solution for the CO displacement experiments at room temperature. Special caution has been taken to avoid the presence of O$_2$ in the CO inlet. All the current densities were calculated by the normalization of the current to the geometric area of the Pt electrode (3.92 mm$^2$). For single crystal electrodes, the active area is equal to the geometrical area.

**Synthesis of Au NPs**. The synthesis of Au nanoparticles was carried out using the Turkevich-Frens citrate reduction method[40–42]. In a round bottomed flask, 2.4 mL 1% HAuCl$_4$ were diluted with 200 mL distilled water (Milli-Q®, 18.2 M$\Omega$). The resulting solution was heated under vigorous stirring until boiling. 1.5 mL 1% sodium citrate were immediately added to the solution of HAuCl$_4$. The dispersion

of nanoparticles was stirred for another 20 minutes and then allowed to cool overnight at room temperature.

**Synthesis of SHINs**. The synthesis of SiO$_2$-coated Au nanoparticles followed the protocol described by Tian et al. [43]. In all, 30 mL of gold nanoparticles (0.07 nM) were placed in a round-bottomed flask under strong stirring. 400 µL of 1 mM APTMS solution were added dropwise and left under vigorous stirring for 15 minutes. Then, 3.2 mL of 0.54% sodium silicate solution (Honeywell) were added and allowed to stir for another 3 minutes. After 3 min, the sample was immersed into a 98 °C oil bath under stirring for 17 min. Finally, the SHINs were quickly cooled down in an ice bath, centrifuged three times, and diluted with pure water. The SHINs were deposited by drop-casting onto Pt (311) single-crystal and dried under Ar for in situ Raman measurements[44,45].

**In situ Raman measurements**. Raman spectra were recorded with an NRS-5100 (Jasco) Raman spectrometer integrated with a confocal microscope. The spectra were obtained by excitation with a 17 mW He–Ne laser with a wavelength of 632.8 nm. Raman was calibrated vs. the 520 cm$^{-1}$ peak of Si with a resolution of 1.0 cm$^{-1}$. For Raman pinhole tests, 5 µL of SiO$_2$-coated SHINs (~0.9 nM) were deposited onto a silicon wafer (Si (100), Agar Scientific) by drop casting. Then, 2 µL of a 10 mM pyridine (99.8%, Sigma-Aldrich) solution were dropped on top of the deposited SHINs (Fig. S5). Enhancement tests were similarly performed but using a gold wafer (Au (111), Platypus Technologies) instead (Fig. S6).

**The cleaning process for the SHINs on the Pt(311) surface**. In order to clean the SHINs, a potential of −2.0 V vs RHE was applied to the Pt(311) electrode after its modification with the SHINs in a 0.1 M HClO$_4$ solution for about 1–2 min and the electrolyte changed prior to any experiments. At this potential value, the hydrogen evolution reaction (HER) induces the desorption and diffusion into the solution of the possible impurities that might remain on the SHINs after the synthesis. Then, the electrode surface was washed carefully with ultrapure water and the solution was changed. The comparison of the cyclic voltammograms before and after the cleaning procedure can be found in the Supplementary information (Fig. S7). Finally, after the cleaning process, the electrode was transferred to another clean Raman cell for the Raman tests.

## Data availability

The data for Figs. 1–4 and supplementary figs. 1, 2, 5, and 6 have been deposited in the figshare repository (http://figshare.com) under accession code (https://doi.org/10.6084/m9.figshare.19583794.v1). All relevant data that support the findings of this study are available from the authors upon reasonable request.

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

## Acknowledgements

R.R, E.H, J.M.F and V.C. acknowledge Ministerio de Ciencia e Innovación (Spain) grant number PID2019-105653GB-I00 and FJC2018-038607-I and Generalitat Valenciana (Spain) grant number PROMETEO/2020/063. J.F.V. acknowledges PhD funding from UKRI Engineering and Physical Sciences Research Council (EPSRC). L.J.H. acknowledges support from Faraday Institution degradation project (FIRG001).

## Author contributions

J.M.F. conceived the project, and R.R. and E.H. planned the experiments. R.R. conducted the electrochemical experiments. F.J.V.I. assisted with setting up the Raman for the in-situ experiments. J.F.V. carried out the in-situ Raman experiments. R.R. and J.F.V. wrote the initial version of the manuscript and supporting information. R.R., J.F.V., F.J.V.I., L.J.H., G.A.A., V.C., E.H., and J.M.F. discussed the results and implications and commented on the manuscript at all stages.

## Competing interests

The authors declare no competing interests.
