## [Peer Review File · Nature Communications]

REVIEWER COMMENTS

Reviewer #1 (Remarks to the Author):

This paper utilized CO displacement experiments and Raman spectroscopy to demonstrate that OH is adsorbed at more negative potentials on the low coordinated Pt. Since the Pt catalyst has shown excellent performance as an electrocatalyst in reactions considering the adsorption of OH species, so the results of this research are interesting and well-organized. Here are some minor issues that need to be addressed before accepting the paper.

Comments:

1. For a better understanding, the authors should provide the full CV profiles before and after the CO displacement experiment for Figure 2b in supporting information.
2. As the article relies on the numerical values of the measured charge and current density, the detailed calculations should be provided in the supporting information so that broader readers can better understand the concept.
3. Is there a reference to the CV profile of the Pt(311) surface?
4. There is no mention about the active surface area and dimensional property of the working electrode. How did the authors calculate the current density?
5. CO displacement experiment is very sensitive to temperature. What was the temperature of the experiment performed?
6. While reviewing this paper, this reviewer began to wonder what happens to the OH adsorption for the low coordinated Pt in alkaline media. Any comments?

Reviewer #2 (Remarks to the Author):

The authors report the work of "Investigating the adsorbed species on Pt steps at low potentials: Is there only hydrogen adsorbed?". This is a systematic and important work, which seems to deserve the publication in more specialized electrochemical journals.

Among the many issues, the reviewer points to:

1. The finding of Pt-OH adsorbed at low overpotentials seems to be reported at J. Am. Chem. Soc. 2020, 142, 715–719 (Figure 2). Please compare this work with yours and make appropriate corrections.
2. Please take the Pt(211) in consideration as well.
3. Take appropriate quotation in your paper for some statement as accurate as possible. For example, the attribution of peak location in Figure S1.

Reviewer #3 (Remarks to the Author):

At the outset, I confess that I have not been in favor of the adsorption of OH_{ad} on Pt surfaces within the Hupd potential region (< 0.4 V vs RHE) in aqueous solution, the major argument of this article by Rizo et al. I am not completely convinced by this article on this argument as well. Despite my stand, I recommend publishing this article at Nature Communication with minor revisions to address few of concerns elaborated below. The article is solid, logical, and well written. In particular, the in situ EC-SHINERS on the Pt(311) surface (Figure 4) is highly appreciated since it is valuable and technically challenging owing to the lack of surface enhancement effects. Below please find my few concerns regarding the interpretation of the data.

(1) One of the experimental evidence for the presence of OH_{ad} on Pt surfaces below 0.4 V is given by the CO displacement experiment with the results displayed in Figure 2b. The authors argued that the negative current is indicative of the presence of OH_{ad} according to the equation $\text{Pt-OH} + e^- + \text{CO} \rightarrow \text{Pt-CO} + \text{OH}^-$. However, strictly speaking, the negative current is essentially indicative of adsorbed species that replacement of it by CO involves electron transfer to the Pt surface. Although the authors preclude the anions, other possibilities such as water with the O bonded with Pt is not impossible. Partial charge transfer between the adsorbed water and Pt electrode has been proposed. The same concern also applies for the interpretation of the experimental data displayed in Figure 3.

(2) The In situ EC-SHINERS spectra displayed in Figure 4 are of high quality. Very impressive. I am not sure why O₂ was observed since there is no source for O₂. More importantly, the emergence of the peak around 861 cm⁻¹ was assigned to Pt-OH and regarded as experimental evidence for the presence of Pt-OH. Is this assignment supported by DFT? Nowadays the DFT can give a very accurate estimation of the wavelength of certain bonds with certain binding configurations. In addition, how does this peak evolve as the potential go up to 0.9 V? Since it is known that OH will adsorb on Pt at elevated potentials and therefore the signals can be used as the reference of OH_{ad} with or without the nearest O neighbors. Essentially the question is whether the peak can be unambiguously assigned to Pt-OH_{ad}.

(3) The final question is more general. If the OH is present on the Pt surface below 0.4 V and responsible for the sharp Hupd peak of Pt by exchanging with H_{ad} as proposed by Koper and his collaborators, what is the source of the OH in acidic electrolyte in which the stepped Pt also gives the sharp Hupd peak as in alkaline? The seemingly only possible source is the water. So to replace the H_{ad}, the Pt needs to break the

water first releasing one proton and then the OH goes to the surface replacing the Had: $\text{Pt-H} + \text{H}_2\text{O} \rightarrow \text{Pt-OH} + 2\text{H}^+ + 2\text{e}^-$, which involves two electron transfer that is not really the case. In addition, if this is the case, it implies that Pt is actually covered by quite some OHad (small amount of OHad will not give such high Hupd peaks) at the double-layer region (~ 0.5 V), which is also against the common sense. These are the major reasons I don't believe the presence of OH on stepped Pt surface at low potentials.

Reviewers' comments to Authors:

Reviewer #1:

This paper utilized CO displacement experiments and Raman spectroscopy to demonstrate that OH is adsorbed at more negative potentials on the low coordinated Pt. Since the Pt catalyst has shown excellent performance as an electrocatalyst in reactions considering the adsorption of OH species, so the results of this research are interesting and well-organized. Here are some minor issues that need to be addressed before accepting the paper.

Our reply: We thank the reviewer for their positive comments about the article.

Comments:

1. For a better understanding, the authors should provide the full CV profiles before and after the CO displacement experiment for Figure 2b in supporting information.

Our reply: Following the reviewer's suggestion, we have added Figure S1 which displays the comparison of the CVs for the Pt(311) surface before and after the CO-displacement experiment.

Figure S1. Cyclic voltammety for Pt(311) in 0.1 M HClO₄ solution before and after the CO displacement experiments.

As shown, the CV remains unperturbed after the experiment.

The information has been added to the manuscript:

Voltammograms recorded before and after the CO displacement experiment were identical (Figure S1) which assures that the surface structure of the electrode is maintained during the experiment.

2. As the article relies on the numerical values of the measured charge and current density, the detailed calculations should be provided in the supporting information so that broader readers can better understand the concept.

Our reply: The following text has been added to the supporting information, as suggested by the reviewer:

Detailed experimental procedure to calculate total charge curves.

Voltammetric charges can be obtained from the integration of the voltammogram as indicated in equation (3) of the manuscript. In this equation, $q(E^*)$ represents the integration constant required to extract absolute charge values from the relative values or charge increments resulting from the integration of the voltammogram between two given limits. This value can be obtained from the displaced charge as indicated in equation (2). To avoid the influence of a possible small offset current either due to a miscalibration of the potentiostat or to the effect of traces of oxygen, the average of the positive and negative current is used in the integration of equation (3).

For Pt(111), the displaced charge at 0.10 V stands for $142 \mu\text{C cm}^{-2}$, and, according to equation (2), $q(0.1 \text{ V}) = -142 \mu\text{C cm}^{-2}$. In combination with the charges obtained from the integration of the voltammogram, the total charge curve shown in red in figure 1 is obtained, resulting in a potential of zero total charge of 0.34 V. In the double layer region, i.e., at 0.51 V, the total charge amounts to $25 \mu\text{C cm}^{-2}$. Since no adsorption processes take place in this potential region, this charge value can be attributed to a purely capacitive (ionic) charge. In contrast, the charge at 0.10 V contains two contributions, namely the true free ionic charge and the charge associated with hydrogen adsorption, according to $q = \sigma_M - F\Gamma_H$, where σ_M is the true electronic charge on the metal and Γ_H is the hydrogen surface excess. According to this relationship, it becomes necessary to differentiate between total charge, q , and free charge, σ_M .

For Pt(311), the displaced charge at 0.10 V and 0.40 V amounts to $90 \mu\text{C cm}^{-2}$ and $-39 \mu\text{C cm}^{-2}$, respectively, while the voltammetric charge integrated between 0.10 and 0.45 V amounts to $130 \mu\text{C cm}^{-2}$. With these values, it can be verified that, within the experimental errors:

$$\int_{0.1 \text{ V}}^{0.40 \text{ V}} \frac{j}{v} dE = q_{dis} (0.1 \text{ V}) - q_{dis} (0.40 \text{ V})$$

Figure 2 shows the excellent agreement between the CO displacement charges (red dots) and the total charges obtained from the voltammetric integration using the displaced charge at 0.1 V as the integration constant (black curve).

Calculation of theoretical charges for the stepped surfaces.

For the surfaces within the series Pt(S)[n(111)×(100)], whose Miller indexes are Pt(n+1,n-1,n-1) the length of the unit cell projected on the plane of the terrace, as shown in figure S4, is:

$$L = \left(n - \frac{1}{3} \right) d \frac{\sqrt{3}}{2}$$

and the projected area of the unit cell is:

$$S' = \left(n - \frac{1}{3} \right) d^2 \frac{\sqrt{3}}{2}$$

Taking into account that there is only a single step atom in each unit cell, the density of steps per unit area can be calculated as:

$$N = \frac{1}{S'} = \frac{2}{d^2 \sqrt{3}} \frac{1}{n - \frac{1}{3}}$$

and the charge corresponding to the exchange of one electron per step site is then:

$$q_{(n+1,n-1,n-1)}^{step} = \frac{e}{S'} = \frac{2e}{d^2 \sqrt{3}} \frac{1}{n - \frac{1}{3}}$$

This charge is projected on the plane of the (111) terrace, which is tilted with respect to the considered surface. The true area, S , and the projected area, S' , are related according

to the cosine of the angle α between the surface (n+1,n-1,n-1) and the (111) plane according to:

$$S' = S \cos(\alpha) = S \frac{3n-1}{\sqrt{9n^2-6n+9}}$$

Thus, the charge corresponding to the exchange of one electron per step site (equation (6) of the manuscript) can be written as:

$$q_{(n+1,n-1,n-1)}^{step} = \frac{e}{S} = \frac{2e}{d^2\sqrt{3}} \frac{1}{n-\frac{1}{3}} \cos(\alpha) = \frac{q_{Pt(111)}}{n-\frac{1}{3}} \cos(\alpha)$$

In this equation, the term $\frac{2e}{d^2\sqrt{3}}$ is the charge associated with the exchange of one electron per unit cell on the (111) surface, $q_{Pt(111)}$.

Figure S7. Hard-sphere model of the Pt(n+1,n-1,n-1) surfaces.

3. Is there a reference to the CV profile of the Pt(311) surface?

Our reply: As requested by the reviewer, the following references to the Pt(311) CV have been added to the manuscript:

25. Dong, J.-C., Su, M., Briega-Martos, V., Li, L., Le, J.-B., Radjenovic, P., Zhou, X.-S., Feliu, J. M., Tian, Z.-Q. & Li, J.-F. Direct in situ Raman spectroscopic evidence of oxygen

reduction reaction intermediates at high-index Pt (hkl) surfaces. *J. Am. Chem. Soc.* **142**, 715–719 (2019).

26. Nakahara, A., Nakamura, M., Sumitani, K., Sakata, O. & Hoshi, N. In situ surface X-ray scattering of stepped surface of platinum: Pt (311). *Langmuir* **23**, 10879–10882 (2007).

27. Climent, V. & Feliu, J. M. Surface electrochemistry with Pt single-crystal electrodes. *Nanopatterned and Nanoparticle-Modified Electrodes* 1–57 (2017).

4. There is no mention about the active surface area and dimensional property of the working electrode. How did the authors calculate the current density?

Our reply: The current density was calculated by the normalization of the current to the geometric area of the bulk Pt electrode (3.92 mm²). This information has been included in the experimental section. It should be highlighted that for single crystal electrodes, the active area is equal to the geometrical area.

5. CO displacement experiment is very sensitive to temperature. What was the temperature of the experiment performed?

Our reply: All the CO displacement experiments were performed at room temperature with a small oscillation in the temperature (293-298 K). The CO displacement experiment is providing the charge difference between the final state (the CO-covered surface) and the initial state (the clean surface). Although the charge for those states is temperature dependent, the observed modifications of the charge in the temperature range between 293 and 298 K (Garcia-Araez, N.; Climent, V.; Feliu, J. M., *J. Solid State Electrochem.* 2008, 12, 387-398) are almost negligible. The effect of the small temperature variations in this range is well below the accuracy of the measurements ($\pm 2 \mu\text{C cm}^{-2}$).

6. While reviewing this paper, this reviewer began to wonder what happens to the OH adsorption for the low coordinated Pt in alkaline media. Any comments?

Our reply: We appreciate the reviewer's feedback. Indeed, we are already working to answer this question. In this sense, we have performed some preliminary experiments which reveal also the adsorption of OH at the steps in alkaline media, regardless of the geometry of the step:

Cyclic voltammogram (black line) recorded at a scan rate of 50 mV s^{-1} and CO displaced charge at different potentials (red dots) on Pt(221) surface.

As can be observed, the sign of the charge displaced at potentials before and after the step changes as a consequence of the adsorption of OH on the step. Our next goal is the study of the correlations between OH adsorbed on (100) and (110) steps and the pH of the solution in the absence of specific adsorption of anions.

Reviewer #2

The authors report the work of “Investigating the adsorbed species on Pt steps at low potentials: Is there only hydrogen adsorbed?”. This is a systematic and important work, which seems to deserve the publication in more specialized electrochemical journals. Among the many issues, the reviewer points to:

Our reply: We appreciate the reviewer’s feedback but we strongly believe that the findings shown here are sufficiently relevant to deserve being published in Nature communication. Pt is the most employed metal in catalysis and particularly OH adsorbed species are considered the main intermediate in most of the electrochemical reactions of interest. Thus, identifying the presence of adsorbed OH at low potentials is of paramount importance in the field of energy and environmental science. This requires a systematic study that has never been done before.

Moreover, other fundamental electrochemistry contributions of relevance have been already published in non-specialized electrochemical journals with great acceptance:

Huang, Y. F., Kooyman, P. J., & Koper, M. (2016). Intermediate stages of electrochemical oxidation of single-crystalline platinum revealed by in situ Raman spectroscopy. *Nature communications*, 7(1), 1-7.

Dong, J. C., Su, M., Briega-Martos, V., Li, L., Le, J. B., Radjenovic, P., ... & Li, J. F. (2019). Direct in situ Raman spectroscopic evidence of oxygen reduction reaction intermediates at high-index Pt (hkl) surfaces. *Journal of the American Chemical Society*, 142(2), 715-719.

There is no doubt that electrochemistry is an increasingly important field of science with a growing presence in many technological applications that deserves more significant visibility in general audience scientific journals.

Among the many issues, the reviewer point to:

1. The finding of Pt-OH adsorbed at low overpotentials seems to be reported at J. Am. Chem. Soc. 2020, 142, 715–719 (Figure 2). Please compare this work with yours and make appropriate corrections.

Our reply: After revising the mentioned article and following the reviewers' suggestions, we have added some corrections and rephrased all the Raman experiment interpretations, based on the article mentioned by the reviewer and other important contributions, that does not alter the main conclusion of the article but help to improve the quality of the discussion.

The Raman discussion now reads as follows:

As expected from the CO displacement experiments displayed in **Figure 2**, electrochemical SHINERS results (**Figure 4**) show no significant Raman peaks at potentials below 0.25 V other than the CaF₂ band from the Raman window, suggesting that only H_{ads} is present on the surface at these potentials. Above 0.25 V, new Raman bands arise at 255, 860, 933, 990, and 1138 cm⁻¹. Despite purging the measurement cell and electrolyte with argon, trace O₂ cannot be fully discounted, in this regard, based on previous DFT studies on Pt(111) surfaces, the band at 990 cm⁻¹ can be correlated to O₂ species bonded to the Pt(311) surface in the presence of OH, corresponding to $\nu_{\text{Pt-O-O}}$ for the [Pt(311)OH₂-O₂]⁺ system³⁴. Moreover, DFT calculations showed that, while the Pt-OH bending mode δ_{PtOH} on Pt(100) sites is around 881 cm⁻¹, the presence of an atomic oxygen nearby causes a shift towards higher wavenumbers in the Pt-OH vibration due to the constructive role that the oxygen atom plays in bending the H atom.³⁴ Thus, the signal at 860 cm⁻¹ is assigned to Pt-OH bending vibrations from OH adsorbed onto (100) steps near Pt atoms not coordinated to atomic oxygen while the signal at 1138 cm⁻¹ can be ascribed to OH adsorbed on Pt atoms surrounded by Pt sites bonded to oxygen. The small deviations in wavenumber

position concerning the theoretical values can be attributed to the effect of energy perturbation due to the presence of the step on the OH bending mode.

The intensity of the bands at 860, 933, and 1138 cm^{-1} increase with the applied potential and then disappear above 0.65 V. The potential dependence of the band intensity and the correlation of the onset potential with the voltammetric signal for the steps (0.25 V) reveal that these contributions must be attributed to adsorbed OH, generated by water dissociation on the steps and not to oxygenated products formed from the reduction of the traces of oxygen in the electrolyte^{6,25}. If that were the case, and considering that the onset potential for oxygen reduction reaction (ORR) is around 0.9 V for Pt(111) surface, these bands should increase by decreasing the applied potential with no dependence on the peak for the step. Further confirmation of the presence of OH adsorbed at the step is the band at 933 cm^{-1} , assigned to the symmetric stretching mode of the perchlorate ion ($\nu_{s(\text{ClO}_4^-)}$). Koper et al. proposed that the formation of OH on Pt (111) is associated with a specific interaction of ClO_4^- with the OH_{ads} layer, which gives rise to the band in the spectra³⁴:

The onset for the band at 933 cm^{-1} is 0.30 V (Figure 4) and becomes more intense at 0.35 V, coinciding with the voltammetric peak for the steps, which suggest that, in analogy with the findings of Koper et al., the interaction of ClO_4^- with the OH_{ads} on the steps is responsible for this emerging band, which disappears by increasing the potential above 0.65 (Figure S2). This type of specific interaction between the Pt-OH surface and perchlorate anion is found to inhibit the ORR suggesting a direct relationship between surface and reactivity.³⁵ Finally, the band at 255 cm^{-1} shows an increasing tendency with the applied potential within the whole potential range. This feature is not easily explained by either of the oxygenated adsorbates described below. Koper et al, assigned bands at similar energies to a superoxide species formed at potentials above 1 V for Pt(111) surface. However, the oxidation of the surface in our working potential range is discarded. Another possibility involves the formation of superoxide species from the reduction of possible traces of oxygen in the solution, however, an increase in this signal with the potential in the opposite direction would be expected. Therefore, the signal at 255 cm^{-1} may be tentatively attributed to a vibration of the Si-O bond³⁶ from the SiO_2 -coated SHINERS employed for the *in situ* Raman experiments, which may experience structural changes induced by the effect of the electric field caused by applying positive potentials. Proposed band assignment and adsorption mechanisms discussed above for potentials above 0.25 V for Pt (111) are shown in **Figure S3**.

The Raman figure has also changed in consequence showing the Raman spectra at two additional potential values to help support our interpretation

Figure 4. Potential dependent *in situ* SHINERS spectra of Pt(311) electrode in 0.1 M HClO₄ electrolyte purged with Ar.

2. Please take the Pt(211) in consideration as well.

Our reply: We appreciate the reviewer's feedback. However, although we strongly believe that a deep study about the correlation between OH adsorption on the steps and the surface step density is needed, we consider that including the study for any other surface would only complicate the reading and understanding of this work. Thus, we think that it is more

appropriate to include the study for the Pt(211) together with the analysis of other surfaces in a subsequent study in which we hope to obtain such correlations that we cannot obtain just by the study of two surfaces.

This is indeed a work that we have already begun. Preliminary studies, as shown below, reveal that OH is adsorbed on the step, regardless of the surface step density, and thus this is not a particular Pt(311) case. As can be observed in the figure below, negative CO-displacement charges are also found for Pt(755) surface.

a) Cyclic voltammogram (black line) surface recorded at a scan rate of 50 mV s^{-1} and CO displacement charge at different potentials (red dots) on Pt(755), b) current-transient curves in the step (0.25 V) and after the peak for the step (0.28V) on the corresponding surface.

3. Take appropriate quotation in your paper for some statement as accurate as possible. For example, the attribution of peak location in Figure S1.

Our reply: Following the reviewer's suggestion, we have doubled checked the article to correct some statements for the Raman interpretation and discussion. Previous Figure S1 (now Figure S3) has now been updated with the band assignment discussed in the paper and is presented as follows:

Figure S3. Schematic representation of species adsorbed at potentials above 0.25 V based on *in situ* SHINERS results.

Reviewer #3

At the outset, I confess that I have not been in favor of the adsorption of OH_{ad} on Pt surfaces within the Hupd potential region (< 0.4 V vs RHE) in aqueous solution, the major argument of this article by Rizo et al. I am not completely convinced by this article on this argument as well. Despite my stand, I recommend publishing this article. at Nature Communication with minor revisions to address few of concerns elaborated below. The article is solid, logical, and well written. In particular, the *in situ* EC-SHINERS on the Pt(311) surface (Figure 4) is highly appreciated since it is valuable and technically challenging owing to the lack of surface enhancement effects. Below please find my few concerns regarding the interpretation of the data.

(1) One of the experimental evidence for the presence of OH_{ad} on Pt surfaces below 0.4 V is given by the CO displacement experiment with the results displayed in Figure 2b. The authors argued that the negative current is indicative of the presence of OH_{ad} according to the equation $\text{Pt-OH} + \text{e}^- + \text{CO} \rightarrow \text{Pt-CO} + \text{OH}^-$. However, strictly speaking, the negative current is essentially indicative of adsorbed species that replacement of it by CO involves electron transfer to the Pt surface. Although the authors preclude the anions, other possibilities such as water with the O bonded with Pt is not impossible. Partial charge transfer between the adsorbed water and Pt

electrode has been proposed. The same concern also applies for the interpretation of the experimental data displayed in Figure 3.

Our reply: We consider very interesting the reviewer's feedback regarding the concern about not contemplating the partial charge transfer coming from adsorbed water in the CO displacement charge. Conceptually, the CO displacement experiment is not able to discern any charge reorganization at the interface since only the charge that flows through the external electric circuit is detected during the quenching of the double layer upon CO adsorption. Therefore, while any partial charge transfer between water and platinum would result in a redistribution of the concepts of free charge and charge involved in adsorption processes, the overall charge that is displaced is only that corresponding to the total charge, that is independent of the microscopic situation. The small contribution of charge that would be required to maintain the potentiostatic condition after the displacement of water dipoles would be included in the remaining charge on the CO-covered surface that has been demonstrated to be very small as a consequence of the very small capacitance of such interphase.

To test the validity of this assumption, many experiments have been performed which demonstrate that water dipoles are not detected by CO displacement. The study of the charge displacement at constant potential during CO adsorption on the Pt basal planes with and without specific anion adsorption revealed that there is good agreement between the charge measured by voltammetry in the absence of CO and the charges measured during CO displacement measurements. This is indicative that the latter charges are produced by the displacement of the species at the interface as a result of CO adlayer formations. Furthermore, many experiments have demonstrated the good agreement between displaced charges and the charges calculated from the hard-sphere model or the structures identified with STM. For example, for well-defined Br-Pt(111), I-Pt(111), and/or Br-Pt(100) in which the well-defined adsorption geometry of the halides was studied by STM, the CO displacement charges coincide exactly with the replacement of the halides monolayer by CO with no extra contribution due to the water dipoles adsorbed on the surface. The information described can be found in the following references:

Clavilier, J., Albalat, R., Gomez, R., Orts, J. M., Feliu, J. M., & Aldaz, A. (1992). Study of the charge displacement at constant potential during CO adsorption on Pt (110) and Pt (111) electrodes in contact with a perchloric acid solution. *Journal of Electroanalytical Chemistry*, 330(1-2), 489-497.

Feliu, J. M., Orts, J. M., Gomez, R., Aldaz, A., & Clavilier, J. (1994). New information on the unusual adsorption states of Pt (111) in sulphuric acid solutions from potentiostatic adsorbate replacement by CO. *Journal of Electroanalytical Chemistry*, 372(1-2), 265-268.

Clavilier, J., Albalat, R., Gómez, R., Orts, J. M., & Feliu, J. M. (1993). Displacement of adsorbed iodine on platinum single-crystal electrodes by irreversible adsorption of CO at controlled potential. *Journal of Electroanalytical Chemistry*, 360(1-2), 325-335.

Orts, J. M., Gomez, R., Feliu, J. M., Aldaz, A., & Clavilier, J. (1996). Nature of Br adlayers on Pt (111) single-crystal surfaces. Voltammetric, charge displacement, and ex situ STM experiments. *The Journal of Physical Chemistry*, 100(6), 2334-2344.

(2) The In situ EC-SHINERS spectra displayed in Figure 4 are of high quality. Very impressive. I am not sure why O₂ was observed since there is no source for O₂. More importantly, the emergence of the peak around 861 cm⁻¹ was assigned to Pt-OH and regarded as experimental evidence for the presence of Pt-OH. Is this assignment supported by DFT? Nowadays the DFT can give a very accurate estimation of the wavelength of certain bonds with certain binding configurations. In addition, how does this peak evolve as the potential go up to 0.9 V? Since it is known that OH will adsorb on Pt at elevated potentials and therefore the signals can be used as the reference of OH_{ad} with or without the nearest O neighbors. Essentially the question is whether the peak can be unambiguously assigned to Pt-OH_{ad}.

Our reply: We thank the reviewer for the good comment about the Raman spectra. Regarding the bands assigned to O₂, we have rephrased the interpretation of the Raman bands, comparing with the literature and considering the presence of traces of oxygen in the solution, which does not alter the main conclusion about the presence of OH adsorption on the steps. The new discussion reads as follows:

As expected from the CO displacement experiments displayed in **Figure 2**, electrochemical SHINERS results (**Figure 4**) show no significant Raman peaks at potentials below 0.25 V other than the CaF₂ band from the Raman window, suggesting that only H_{ads} is present on the surface at these potentials. Above 0.25 V, new Raman bands arise at 255, 860, 933, 990, and 1138 cm⁻¹. Despite purging the measurement cell and electrolyte with argon, trace O₂ cannot be fully discounted, in this regard, based on previous DFT studies on Pt(111) surfaces, the band at 990 cm⁻¹ can be correlated to O₂ species bonded to the Pt(311) surface in the presence of OH, corresponding to $\nu_{\text{Pt-O-O}}$ for the [Pt(311)OH₂-O₂]⁺ system³⁴. Moreover, DFT calculations showed that, while the Pt-OH bending mode δ_{PtOH} on Pt(100) sites is around 881 cm⁻¹, the presence of an atomic oxygen nearby causes a shift towards higher wavenumbers in the Pt-OH vibration due to the constructive role that the oxygen atom plays in bending the H atom.³⁴ Thus, the signal at 860 cm⁻¹ is assigned to Pt-OH bending vibrations from OH adsorbed onto (100) steps near Pt atoms

not coordinated to atomic oxygen while the signal at 1138 cm^{-1} can be ascribed to OH adsorbed on Pt atoms surrounded by Pt sites bonded to oxygen. The small deviations in wavenumber position concerning the theoretical values can be attributed to the effect of energy perturbation due to the presence of the step on the OH bending mode.

The intensity of the bands at 860, 933, and 1138 cm^{-1} increase with the applied potential and then disappear above 0.65 V. The potential dependence of the band intensity and the correlation of the onset potential with the voltammetric signal for the steps (0.25 V) reveal that these contributions must be attributed to adsorbed OH, generated by water dissociation on the steps and not to oxygenated products formed from the reduction of the traces of oxygen in the electrolyte^{6,25}. If that were the case, and considering that the onset potential for oxygen reduction reaction (ORR) is around 0.9 V for Pt(311) surface, these bands should increase by decreasing the applied potential with no dependence on the peak for the step. Further confirmation of the presence of OH adsorbed at the step is the band at 933 cm^{-1} , assigned to the symmetric stretching mode of the perchlorate ion ($\nu_{s(\text{ClO}_4^-)}$). Koper et al. proposed that the formation of OH on Pt (111) is associated with a specific interaction of ClO_4^- with the OH_{ads} layer, which gives rise to the band in the spectra³⁴:

The onset for the band at 933 cm^{-1} is 0.30 V (Figure 4) and becomes more intense at 0.35 V, coinciding with the voltammetric peak for the steps, which suggest that, in analogy with the findings of Koper et al., the interaction of ClO_4^- with the OH_{ads} on the steps is responsible for this emerging band, which disappears by increasing the potential above 0.65 (Figure S2). This type of specific interaction between the Pt-OH surface and perchlorate anion is found to inhibit the ORR suggesting a direct relationship between surface and reactivity.³⁵ Finally, the band at 255 cm^{-1} shows an increasing tendency with the applied potential within the whole potential range. This feature is not easily explained by either of the oxygenated adsorbates described below. Koper et al, assigned bands at similar energies to a superoxide species formed at potentials above 1 V for Pt(111) surface. However, the oxidation of the surface in our working potential range is discarded. Another possibility involves the formation of superoxide species from the reduction of possible traces of oxygen in the solution, however, an increase in this signal with the potential in the opposite direction would be expected. Therefore, the signal at 255 cm^{-1} may be tentatively attributed to a vibration of the Si-O bond³⁶ from the SiO_2 -coated SHINERS employed for the *in situ* Raman experiments, which may experience structural changes induced by the effect of the electric field caused by applying positive potentials. Proposed band assignment and adsorption mechanisms discussed above for potentials above 0.25 V for Pt (311) are shown in **Figure S3**.

Regarding the band at 860 cm^{-1} , DFT calculations performed by Dong et al. (see reference below) showed that the vibrational frequency of the bending mode of Pt-OH at Pt(100) was located at 875 cm^{-1} , which correlated well with the experimental results. The small deviations concerning the theoretical value should be attributed to the effect of energy perturbation due to the presence of the step on the OH bending mode.

Dong, J.-C., Zhang, X.-G., Briega-Martos, V., Jin, X., Yang, J., Chen, S., Yang, Z.-L., Wu, D.-Y., Feliu, J. M. & Williams, C. T. In situ Raman spectroscopic evidence for oxygen reduction reaction intermediates at platinum single-crystal surfaces. *Nat. Energy* **4**, 60–67 (2019).

Unfortunately, the evolution and nature of this band cannot be determined by recording the CV up to 0.9 V, as suggested by the reviewer, since voltammetric currents at potentials above the double layer are very low for the Pt(311) surface (see the CV below)

Figure S2. Cyclic voltammetry for Pt(311) in 0.1 M HClO₄ solution up to 0.85 V. It is possible to observe that, unlike Pt(111) and long terrace surfaces, Pt(311) shows almost no OH adsorption states at potential values more positive than the peaks attributed to the (100) steps.

(3) The final question is more general. If the OH is present on the Pt surface below 0.4 V and responsible for the sharp Hupd peak of Pt by exchanging with Had as proposed by Koper and his collaborators, what is the source of the OH in acidic electrolyte in which the stepped Pt also

gives the sharp Hupd peak as in alkaline? The seemingly only possible source is the water. So to replace the Had, the Pt needs to break the water first releasing one proton and then the OH goes to the surface replacing the Had: $\text{Pt-H} + \text{H}_2\text{O} \rightarrow \text{Pt-OH} + 2\text{H}^+ + 2\text{e}^-$, which involves two electron transfer that is not really the case. In addition, if this is the case, it implies that Pt is actually covered by quite some OHad (small amount of OHad will not give such high Hupd peaks) at the double-layer region (~ 0.5 V), which is also against the common sense. These are the major reasons I don't believe the presence of OH on stepped Pt surface at low potentials.

Our reply: As mentioned by the reviewer, the water is the only possible source of OH⁻ and H⁺ which contributes to the charge below 0.4 V in the cyclic voltammetry. However, even if the apparent reaction involves 2 electrons ($\text{Pt-H} + \text{H}_2\text{O} \rightarrow \text{Pt-OH} + 2\text{H}^+ + 2\text{e}^-$), the stoichiometry of is not correct. The maximum coverage of hydrogen on the step, according to our measurements, is ca. 0.84 and that of OH is 0.24. Thus, the reaction can be written as:

Due to repulsive interactions between adsorbed hydrogen, the coverage is smaller than 1. The most stable configuration of adsorbed OH on the step is the bridge-bonded mode, that is, OH is bonded to two contiguous atoms on the step (C. Busó-Rogero, E. Herrero, J. Bandlow, A. Comas-Vives, T. Jacob, Phys. Chem. Chem. Phys. 2013, 15, 18671). The measured value for the OH coverage on the step is smaller than that measured for the Pt(111) or Pt(100) electrodes (ca. 0.33 and 0.5 respectively).

REVIEWERS' COMMENTS

Reviewer #1 (Remarks to the Author):

The manuscript has been revised by fully reflecting the reviewers' comments. Therefore, the current version looks publishable as is.

Reviewer #3 (Remarks to the Author):

The authors addressed most of my questions except for two small concerns.

(1) Despite the technical challenges the authors explained, it will be more convincing that the evolution and nature of the Raman band of OH can be determined by recording the CV up to 0.9 V for any Pt facet, because it is known for sure that OHad is present.

(2) The equation $\text{Pt-H}0.84 + 0.24\text{H}_2\text{O} \rightarrow \text{Pt-OH}0.24 + 1.08\text{H}^+ + 1.08 \text{e}^-$ given by the authors is not based on per site as I previously wrote (Koper et al gave the same one (eq. 2; Journal of Catalysis 367 (2018) 332–337)), but the whole Pt surface taking the particle coverage under consideration, thus giving a very different charge transfer number. The equation given by the authors is similar to the Eq. 10 reported in Catalysis Today 202 (2013) 105– 11, but replaced by another one (Eq. 11) owing to its “artificial looking”. Charge transfer per site or per Had is more fundamental since it is directly related to the pH-dependent shift of the Hupd peak caused by this reaction.

Reviewers' comments to Authors:

Reviewer #1

The manuscript has been revised by fully reflecting the reviewers' comments. Therefore, the current version looks publishable as is.

Our reply: We thank the reviewer for their positive comment about our revision of the article.

Reviewer #3

The authors addressed most of my questions except for two small concerns. (1) Despite the technical challenges the authors explained, it will be more convincing that the evolution and nature of the Raman band of OH can be determined by recording the CV up to 0.9 V for any Pt facet, because it is known for sure that OHad is present.

Our reply: The evolution and nature of the OH Raman band can be confirmed by recording the CV up to 0.9 V for surfaces with long (111) terraces. However, for the Pt(311) surface, considered the turning point between (111) and (100) terraces (2 atoms in the terrace divided by a monoatomic step), the OH adsorption process takes place in a different potential region. In analogy with surface containing (100) terraces, the increase in current observed at potentials higher than 0.8 V (Figure 1) for the Pt(311) surface is due to surface oxidation rather than OH adsorption¹⁻². This latter process should be avoided since it leads to surface order damage. Thus, in this case, the OH adsorption process takes place in the potential range between 0.2 and 0.45 V, as described in the article, and not at high potentials. Therefore, the nature of the OH band cannot be confirmed by recording the CV up to 0.9 V.

Figure 1. Cyclic voltammetry for Pt(311) in 0.1 M HClO₄ solution up to 0.85 V.

¹K. Domke, E. Herrero, A. Rodes, J.M. Feliu, J. Electroanal. Chem., 552 (2003), p.115

²A. Rodes, K. El Achi, M.A. Zamakhchari, J. Clavilier, J. Electroanal. Chem., 284 (1990), p. 245

(2) The equation $\text{Pt-H}_{0.84} + 0.24\text{H}_2\text{O} \rightarrow \text{Pt-OH}_{0.24} + 1.08\text{H}^+ + 1.08 \text{e}^-$ given by the authors is not based on per site as I previously wrote (Koper et al gave the same one (eq. 2; Journal of Catalysis 367 (2018) 332–337)), but the whole Pt surface taking the particle coverage under consideration, thus giving a very different charge transfer number. The equation given by the authors is similar to the Eq. 10 reported in Catalysis Today 202 (2013) 105– 11, but replaced by another one (Eq. 11) owing to its “artificial looking”. Charge transfer per site or per Had is more fundamental since it is directly related to the pH-dependent shift of the Hupd peak caused by this reaction.

Our reply: We are aware that the stoichiometry proposed to answer the reviewer is just a tentative stoichiometry to explain that in the OH adsorption process on the Pt surface less than two electrons are involved per Pt surface site (which is the actual value measures) in the two-electrons transfer process suggested by the reviewer ($\text{Pt-H} + \text{H}_2\text{O} \rightarrow \text{Pt-OH} + 2\text{H}^+ + 2\text{e}^-$) during the first round of revisions. However, the fast dynamic of this step makes it difficult to envisage

Universitat d'Alacant
Universidad de Alicante

Instituto Universitario de Electroquímica
Institut Universitari d'Electroquímica

Tel. 96 590 9814
Campus de Sant Vicent del Raspeig
Apt. 99, E-03080 Alacant
E-mail: iue@ua.es

the exact equation for this process. That is the reason why this explanation is not included in the manuscript, which, in any case, perturbs the main conclusion of the article.